# The In Vitro Contractile Response of Canine Pregnant Myometrium to Oxytocin and Denaverine Hydrochloride

**DOI:** 10.3390/biology12060860

**Published:** 2023-06-15

**Authors:** Carolin Jungmann, Sophie-Charlotte Pyzik, Eva-Maria Packeiser, Hanna Körber, Susanne Hoppe, Gemma Mazzuoli-Weber, Sandra Goericke-Pesch

**Affiliations:** 1Reproductive Unit, Clinic for Small Animals, University of Veterinary Medicine Hannover, 30559 Hannover, Germany; carolin.jungmann@tiho-hannover.de (C.J.); eva-maria.packeiser@tiho-hannover.de (E.-M.P.); hanna.koerber@tiho-hannover.de (H.K.); 2Tierklinik Neustädter Bucht, 23730 Sierksdorf, Germany; 3Institute for Physiology and Cell Biology, University of Veterinary Medicine Hannover, 30173 Hannover, Germanygemma.mazzuoli-weber@tiho-hannover.de (G.M.-W.)

**Keywords:** canine, parturition, dystocia, oxytocin, denaverine hydrochloride, myometrium, contractility

## Abstract

**Simple Summary:**

Conservative treatment of canine dystocia with oxytocin and occasionally denaverine hydrochloride is usually unsuccessful. In this study, we used an in vitro organ bath assay to investigate the potential ecbolic effects of oxytocin and denaverine on pregnant canine myometrium. The circular and longitudinal myometrial layer of canine uterine tissue obtained during emergency Caesarean section were dissected, cut into strips, and mounted in the organ bath. These strips were stimulated with different concentrations of oxytocin and denaverine hydrochloride and the resulting changes in contractility and contraction pattern were evaluated statistically. Our experiments showed different effects on contractility depending on the oxytocin concentrations used. We further identified differences in responses between layers, especially when high doses of oxytocin were applied. Repetitive stimulation with high doses of oxytocin caused the longitudinal layer to stop contracting completely. In contrast, low doses of oxytocin had a better effect on the contractility of both layers and, thus, are recommended for use in clinics. Experiments with denaverine hydrochloride showed no effect on myometrial contractility.

**Abstract:**

In pregnant bitches, the response to oxytocin and denaverine hydrochloride in dystocia management is usually poor. To better understand the effect of both drugs on myometrial contractility, the circular and longitudinal muscle layers were examined in an organ bath. For each layer, three myometrial strips were stimulated twice, each with one of three oxytocin concentrations. The effect of denaverine hydrochloride was studied once in direct combination with oxytocin and alone with subsequent oxytocin administration. Contractions were recorded and evaluated for average amplitude, mean force, area under the curve (AUC), and frequency. Effects of different treatments were analyzed and compared within and between layers. In the circular layer, oxytocin significantly increased amplitude and mean force compared to untreated controls regardless of stimulation cycles or concentrations. In both layers, high oxytocin concentrations caused tonic contractions, while the lowest concentration created regular rhythmic contractions. Longitudinal layer tissue responded to oxytocin with a significantly decreased contractility when stimulated twice, presumably a sign of desensitization. Denaverine hydrochloride neither affected oxytocin induced contractions nor showed a priming effect to subsequent oxytocin. Thus, no benefit of denaverine hydrochloride on myometrial contractility was found in the organ bath. Our results suggest a better efficiency of low-dose oxytocin in canine dystocia management.

## 1. Introduction

Canine pregnancy is maintained by progesterone (P4) solely of luteal origin. Around day 60, P4 drops below a critical threshold level [1,2,3], resulting in changes in the complex interplay of feto–maternal communication. This subsequently initiates the luteolytic cascade, thus, signaling the onset of parturition [3,4,5,6,7]. Luteolysis, decreasing P4, and increasing circulating prostaglandin (PG) F2α concentrations are accompanied by a strong immune and inflammatory response [8,9]. With the omission of the P4-mediated quiescence of the myometrium, the uterus becomes responsive for contractile stimuli, such as endogenous oxytocin [1,10,11]. Additionally, prepartum upregulation of the oxytocin receptor (OXTR) suggests the involvement of oxytocin in prostaglandin signaling [11,12,13].

Knowledge about the underlying physiological endocrine and molecular changes during normal canine parturition has significantly increased. Nonetheless, understanding of the changes during canine dystocia, especially uterine inertia, remains limited. Primary uterine inertia, the lack of functional myometrial contractions at term, is responsible for more than 70% of canine dystocia cases and, thus, the most common maternal cause for labor complications [14,15,16,17]. Failure of luteolysis was suspected to be causative for insufficient myometrial contractions [10,18,19,20]. Nevertheless, it could not be confirmed in our latest study comparing bitches with primary uterine inertia and obstructive dystocia with still strong myometrial contractions [21]. However, elevated progesterone receptor expression in primary uterine inertia might have enhanced the availability of remaining P4 [21]. Whereas expression of elements of the prostaglandin pathway was not altered [22,23], significant differences in contraction-associated proteins between primary uterine inertia and obstructive dystocia were identified. Smooth muscle γ-actin and myosin gene expressions were significantly higher in primary uterine inertia compared to obstructive dystocia [24]. *ROCK1* and *ROCK2*, the two effector kinases of the GTPase RhoA that regulates smooth muscle contractions through calcium sensitization [25,26], were found to be downregulated due to fatiguing prolonged contractions in obstructive dystocia [27].

Medical approaches to attempt treatment of uterine inertia include administering oxytocin as an ecbolic drug. Various oxytocin concentrations have been described with especially recently lower dosages being recommended to effectively increase the quality of myometrial contractions [28] instead of high dosages (5–20 I.U. [29] vs. 0.5–3 I.U. [30]). In dystocia cases with more than two puppies remaining, even lower concentrations (0.25–0.5 I.U.), administered more frequently, were recommended [31]. Some clinicians combine oxytocin with denaverine hydrochloride to create regular uterine contractions. However, failure rates of medical treatment attempts are high (54.3% of bitches) [31]. Consequently, caesarean (C-) section often remains the first choice [32].

The nonapeptide hormone oxytocin is one of the most potent uterotonic factors. Its release from the pituitary gland is triggered by intracervical pressure when a puppy enters the birth canal [33]. The effect of oxytocin is largely mediated via binding to the specific oxytocin receptor (OXTR), which belongs to the family of the G-protein-coupled receptors. Canine serum oxytocin concentrations were already studied in the context of uterine inertia, albeit with contradictory results [18,34,35]. While Bergström et al. [35] described low plasma oxytocin concentrations in bitches diagnosed with primary uterine inertia, in a subsequent study the same research group [18] could not identify any differences between the oxytocin concentrations of bitches with primary uterine inertia and a control group. Recently, our working group found significantly higher *OXTR* gene expression in primary uterine inertia compared to obstructive dystocia [21], supporting the data of Tamminen et al. [36], who also identified slightly higher *OXTR* expression in complete primary uterine inertia compared to obstructive dystocia. Interestingly, we further identified significant differences in OXTR protein expression between the inner circular myometrial layer and the outer longitudinal layer in primary uterine inertia as well as obstructive dystocia, allowing us to hypothesize that there might be a different responsiveness to oxytocin between the two muscle layers [21]. These observations indicate the need for further investigations to study the two myometrial layers separately.

Denaverine hydrochloride was formerly registered for use in dogs but is currently only available for cows. Data on denaverine hydrochloride are limited, not allowing for reliable statements to be made about its mode of action and effectiveness. Based on the mode of action of papaverine, the benzylic acid derivative denaverine is described as a “neurotropic-musculotropic spasmolytic agent”, decreasing the tone of the muscle of the soft birth canal (cervix, vagina, vulva) without being tocolytic [31,37,38,39]. While the neurotropic effect is attributed to competitive inhibition of M-cholinergic receptors, not much is known about the myotropic component [40]. Presumably, the effect thereof is due to inhibition of the calcium uptake of depolarized muscle cells caused by a blockade of chemo-sensitive and slow potential-controlled calcium channels of the cell membrane [40]. In dogs, however, denaverine hydrochloride is postulated “to reactivate stagnating parturitions” [41], contributing to better efficiency and coordination of uterine contractions [37,42]. In addition, in canine uterine inertia, the combined administration of oxytocin and denaverine hydrochloride is recommended; oxytocin should initiate uterine contractions, and denaverine is suggested to compensate prolonged contracture after oxytocin [42]. In fact, these postulates have never been confirmed, indicating the need for detailed investigations into a possible mode of action.

Despite the high failure rates of medical treatment attempts, no studies on canine uterine contraction patterns and contractile responses to ecbolic drugs have been conducted. However, such studies would be crucial for the understanding of canine uterine contractions during parturition. Furthermore, the oxytocin dose–response relationships of the canine myometrium are completely missing, although the clinical dosing recommendations have recently been adjusted [30,43]. Therefore, the aim of this study was to investigate the contraction ability and pattern of canine pregnant myometrium in vitro using an organ bath assay. More precisely, we analyzed the contractile activity of the two myometrial layers separately from bitches presented for a medically indicated C-section. Although connected in vivo by a dense stratum vasculosum, we separated the two myometrial layers for the experiment to assess their contraction behavior individually. The tissues were exposed to different concentrations of oxytocin and denaverine hydrochloride. We hypothesized, based on our previous findings [21], that the response to exogenous oxytocin differs between the two myometrial layers.

## 2. Materials and Methods

Animal experimentation was approved by the respective authorities (Lower Saxony State Office for Consumer Protection and Food Safety (LAVES), permit AZ 20/3360).

### 2.1. Tissue Collection

To assess the uterokinetic properties of oxytocin, interplacental uterine tissue was collected from pregnant bitches around term during medically indicated C-sections with/without ovariohysterectomy (the latter depending on medical indication, or on owner’s request). Bitches were either presented for emergency C-section during parturition or for an elective C-section. Elective C-sections were performed between day 61–63 after ovulation with obvious fetal intestinal motility visualized by ultrasound confirming fetal maturity [44]. Detailed information about all the bitches included in this study is given in Appendix A. Mean body weight of the bitches was 20.84 ± 15.89 kg (range: 2.45–69 kg) and mean age was 5.1 ± 1.7 years (range: 2–9 years). For the experiments in which the effect of oxytocin was studied, tissue was obtained from 19 bitches. Samples of six other bitches were used to investigate the combined effect of denaverine and oxytocin. In order to ensure comparability of the samples, uterine interplacental tissue was always taken from the left uterine horn, opened along the greater curvature between the first and second puppy (except for singleton pregnancies, *n* = 6). In two cases, the incision was performed in the uterine body due to medical reasons. The fresh tissue was immediately transferred into physiological saline solution and stored at 4 °C until further preparation. Time between harvesting of the sample and start of the actual experiment varied between 3 and 25 h depending on the time points of the surgery.

### 2.2. Sample Preparation and Mounting in the Organ Bath

To investigate motility of the myometrial tissue, two organ bath set ups (custom made by the precision mechanic workshop of the Institute for Physiology and Cell Biology, University of Veterinary Medicine Hannover, Foundation, Hannover, Germany) were used, with eight tissue chambers each. The performed method of dissection has previously been successfully described for guinea pig gastrointestinal tissue [45].

Following removal of the endometrium, the two myometrial layers (circular and longitudinal) were carefully dissected using a stereomicroscope (Olympus SZ30 Stereomikroskop, Olympus Deutschland SE & Co. KG, Hamburg, Germany). Throughout the dissection process, the tissue was covered with ice cold, preoxygenated modified Krebs solution composed of (in mmol/L) 1.2 MgCl_2_, 2.5 CaCl_2_, 1.2 NaH_2_PO_4_, 117 NaCl, 25 NaHCO_3_, 11 glucose, and 4.7 KCl, which was exchanged every 10 min. Four strips of 1 cm length and 0.25 cm width aligned with the direction of the muscle fibers were prepared from each myometrial layer. The strips were then mounted one at a time in the eight chambers of the organ bath, filled with 12 mL of preheated 37 °C warm modified Krebs solution. The Krebs solution consisting of (in mmol/L) 1.2 MgCl_2_, 2.5 CaCl_2_, 1.2 NaH_2_PO_4_, 117 NaCl, 20 NaHCO_3_, 11 glucose, and 4.7 KCl, was constantly oxygenated with 95% O_2_ and 5% CO_2_. The buffer temperature was maintained throughout the experiment by warm distilled water, which was passed through a heat exchanger (Haake D1 Heating Circulator, Thermo Fisher Scientific, Waltham, MA, USA) circulating around the chambers in a double Plexiglas wall.

Myometrial strips were connected to an isometric force transducer (Hottinger Brüel & Kjaer GmbH, Darmstadt, Germany) in order to measure myometrial contractions. A resting tension of 2 g (~20 mN) was applied to all strips, simulating the in vivo pre-tension to cause physiological phasic and tonic contractions. The contractility variations measured by the force transducer were converted from mechanical to electrical signals by a measuring transformer (Spider8 PC-Messelektronik, Hottinger Brüel & Kjaer GmbH) and recorded with the software Catman^®^ Easy (Catman© Easy, Version 1.01, Hottinger Brüel & Kjaer GmbH) as previously described [45,46].

### 2.3. Stimulation with Oxytocin

Figure 1 shows a typical experimental run using oxytocin as ecbolic agent. After mounting the tissue strips as described above, the experiment was started with an equilibration period of 60 min. After that, uterokinetic effects were recorded in 20–min time frames (TF), starting with a 20–min period to capture the status quo prior to the first stimulation (TF1). In the following time frame (TF2), the myometrial strips were stimulated with oxytocin (Oxytocin 10 IE/mL Injektionslösung, Serumwerk Bernburg AG, Bernburg, Germany). One strip of each layer served as untreated control. The other three strips per layer were exposed to either 1 nM, 10 nM or 100 nM oxytocin. The concentrations were adapted from the clinically used dosages and comparable experiments from the literature [47]. After each exposure to oxytocin, the myometrial strips were washed twice with Krebs solution to remove potential oxytocin remnants, thereafter allowing them to recover for 20 min (TF 3, 5, 7). Stimulation with oxytocin was repeated three times (TF 2, 4, 6) to investigate the reproducibility of the results. As the effect of oxytocin did not visually differ between the second and third application of oxytocin, we decided to focus on the first and second stimulation cycle in the analysis.

### 2.4. Stimulation with Denaverine

Denaverine hydrochloride (Sensiblex©, denaverine hydrochloride 40 mg/mL, Veyx-Pharma GmbH, Schwarzenborn, Germany) was used in quantities of 1.97 nM and 2.97 nM, extrapolated from the manufacturer’s instructions [41] for in vivo application. As combined use of denaverine and oxytocin was recommended by the manufacturer in the case of uterine inertia, we included this in our experiment. The course of the experiment is illustrated in Figure 2. At the first stimulation (TF2), two strips of each layer were stimulated with 1 nM oxytocin, followed by additional application of either 1.97 nM or 2.97 nM denaverine one min later. The third strip of each layer was stimulated with 1 nM oxytocin only as a control to the combined effect of oxytocin and denaverine. The fourth strip was exposed to 2.97 nM denaverine only to check for an individual effect. In a second stimulation (TF4), strips receiving only 2.97 nM denaverine at the first stimulation (TF2) were stimulated with oxytocin to investigate a possible priming effect of denaverine hydrochloride on oxytocin-induced contractility.

### 2.5. Data Analysis

During the experiments, raw data from all timeframes (TFs) were saved in an Excel file (Microsoft Excel 2016, Microsoft Corporation, Redmond, WA, USA). In every TF (1–5), 10 contractions, if any, were evaluated. The peak of each contraction and the baseline tension before and after the contractions were determined manually, and the difference was used to calculate the amplitude, meaning the maximum force of each contraction. In some cases, the voltage did not drop back down to baseline tension before rising again to a peak. In this case, the contraction was defined as finished when the tension reached below 70% of the tension difference between the baseline tension and the peak. If a contraction was directly followed by a minor one, with less than 0.1 g above the baseline tension, it was not counted as an additional contraction but included in the previous one. Generally, increases in baseline tension below 0.5 g were not considered contractions. The time from the initial increase in tension to the return to baseline tension was determined as the duration of the contraction, and, thus, the mean force per contraction was calculated. Furthermore, the area under the curve (AUC) was used to measure the contractile effects of each agonist concentration and calculated as the integral of the time interval of one contraction. The frequency of contractions in 10 min was counted. Results were transformed from g into mN (amplitude and mean force), or from g·s into mN·s (AUC) by multiplying by 9.8. Finally, individual values for amplitude, mean force, and AUC were averaged for all the contractions examined per TF and used for the following statistical analysis. As 1 nM oxytocin, eliciting relatively equal contractions, was used in all denaverine experiments, the same method of analyzing mean values was retained here.

In some strips exposed to 10–100 nM oxytocin, we observed an initial long-lasting contraction followed by many other very short, weaker contractions instead of steady and equal ones. Consequently, in a second approach, we analyzed only the initial response of the strips to the respective concentrations of oxytocin for the parameters described above.

### 2.6. Statistical Analysis

For all statistical tests and graphical presentations, Graph Pad Prism9 software (GraphPad Software, Inc., La Jolla, CA, USA) was used.

Statistical analysis was performed to analyze the direct effect of oxytocin and denaverine on the parameters of contraction to be measured, including amplitude, contraction mean force, and frequency, as well as the integration of these values to generate mean integral force or AUC.

For the analysis of the oxytocin stimulation, a two-way repeated measurement (RM) ANOVA was used to investigate the effects of the three different oxytocin concentrations (factor “concentration”) and the two stimulation cycles (factor “stimulation cycle”) on the average amplitude/mean force/AUC/frequency of contractions in the two different myometrial layers, respectively. In the case of a missing value due to a loosened knot during the experiment (*n* = 3 stripes), the mixed effects model was applied. The ANOVA/mixed effects model investigated the simple main effects of each independent factor included and checked for interaction between them. When the ANOVA/mixed effects model revealed significant differences, post hoc comparison of the means was performed. Tukey’s multiple comparison test was utilized to investigate the effects of the different concentrations. Additionally, Šídák’s multiple comparisons test was used to compare the first and second stimulation within the respective layer. In the following step, the responses of the two layers (factor “layer”) were compared at both stimulation TFs (factor “stimulation cycle”) for each oxytocin concentration separately. For this, we used a two-way RM ANOVA or a mixed effects model, followed by Šídák’s multiple comparisons test in case of significant differences. If not stated otherwise, all data are presented as mean ± standard deviation. Additionally, a possible influence of the covariant storage time of samples until experiment, age of bitches at C-section, and P4 serum concentration was investigated using SAS Studio (SAS Institute, Cary, NC, USA).

Due to differences in the experimental design of the denaverine study, a different statistical approach was utilized here. For each layer, strips stimulated with 1 nM oxytocin followed by 1.97 nM or 2.97 nM denaverine in the same TF were compared to the control receiving only 1 nM oxytocin. The individual effect of denaverine was assessed, comparing the stimulation (TF2) with the equilibration time frame before (TF1). To investigate the effect of prior denaverine-priming on oxytocin-induced contractility, strips stimulated with oxytocin at only the first stimulation (TF2) were compared to those stimulated with oxytocin (TF 4) after receiving only 2.97 nM denaverine (TF2). A paired *t*-test was used for normally distributed data. Otherwise, the Wilcoxon matched-pairs signed rank test was utilized. To allow a uniform presentation of data, all results are presented as mean ± standard deviation.

All differences were considered to be significant if *p* ≤ 0.05.

## 3. Results

### 3.1. Oxytocin

#### 3.1.1. General Observations

The typical responses of the two layers during oxytocin exposure are shown in Figure 1. A total of 152 tissue strips were used in the 19 organ bath experiments. In 14 strips (11 longitudinal ones and 3 circular ones, obtained from 6 bitches), neither spontaneous activity nor response to oxytocin was observed. Thus, these tissues were excluded from the analysis. In the longitudinal layer, a total of 30 strips and in the circular layer a total of 27 strips showed no spontaneous contractions before stimulation with oxytocin, but a decent response to the ecbolic agent.

Whereas both layers showed regular, rhythmic contractions in response to 1 nM oxytocin (24/34 stripes in the first and 28/34 stripes in the second stimulation), the response to 10 nM and 100 nM was quite different. In the circular layer, the high and medium oxytocin concentration caused a permanent tonic contraction in seven strips from five bitches. In three strips, these contractions lasted up to 20 min, maintaining a relatively constant mean force. Subsequent contractions were regular and shorter. After the wash out, all circular strips maintained uniform regular contractions. Once stimulated, no strip in the circular layer stopped contracting afterwards.

In the longitudinal layer, both high oxytocin concentrations (10 nM and 100 nM) caused only a single strong spastic contraction in 18 of 31 strips. In contrast to the contraction observed in the circular layer, tension peaked following oxytocin application and then slowly declined within 5–10 min until settling at a level above the baseline tension. When applied for the second time, no response to 10 nM and 100 nM oxytocin was identified in 27 of 31 strips from the longitudinal layer.

Covariance analysis showed no significant effect of sample storage time, age of bitches, or serum P4-concentrations on the parameters of amplitude, mean force, AUC, and frequency of contractions.

#### 3.1.2. Amplitude

Two-way RM ANOVA revealed a significant interaction of the independent variables (concentration and stimulation cycle) in the longitudinal (*p* ≤ 0.001) but not in the circular layer.

Simple main effects analysis showed that the used concentrations had a statistically significant effect on the amplitude in the circular layer (*p* ≤ 0.001). All concentrations caused a significant increase in the amplitude of contractions compared to the non-stimulated control regardless of the stimulation time point (first stimulation: *p* ≤ 0.01 each; second stimulation: *p* ≤ 0.001, each; Figure 3a). When comparing the responses of the three concentrations among each other for each stimulation cycle separately, the resulting amplitudes did not differ. According to the main effect analysis, there were no differences when comparing the effects of the three oxytocin concentrations between stimulation cycles in the circular layer.

In the longitudinal layer, simple main effects analysis showed that the oxytocin concentration used (*p* ≤ 0.05) as well as the stimulation cycle (*p* ≤ 0.001) had a statistically significant effect on the amplitude. At the first stimulation, the longitudinal layer responded to all oxytocin concentrations, showing an increase in amplitude compared to the control. However, this increase was only significant for 1 nM (*p* = 0.021; Figure 4a). Surprisingly, when stimulated again with 10 nM and 100 nM, the amplitude of the stimulated strips was significantly lower compared to the first stimulation (*p* ≤ 0.001) and was even lower than in the untreated control group. Consequently, comparison of the three concentrations showed no differen ces at first stimulation, but there was a significantly stronger response to 1 nM compared to 10 nM and 100 nM (*p* ≤ 0.01 for each) at the second stimulation in the longitudinal layer (Figure 4a).

Comparing myometrial layers, two-way RM ANOVA revealed a significant interaction of the respective layer and the stimulation cycle for all three oxytocin concentrations for the amplitude (*p* ≤ 0.01, each). A significant effect of the stimulation cycle on the amplitude was found for 10 nM (*p* ≤ 0.01) and 100 nM (*p* ≤ 0.001), with a significantly reduced amplitude in the longitudinal layer at the second oxytocin application. This explains the observed significant main effect of the factor “layer” (*p* ≤ 0.01, each; Figure 5a). Figure 3(**a**) Amplitude, (**b**) mean force, (**c**) AUC, and (**d**) frequency of contractions of circular myometrial strips stimulated with 1 nM, 10 nM, and 100 nM oxytocin. Responses following stimulation differed significantly from respective untreated controls (## *p* ≤ 0.01, ### *p* ≤ 0.001). The results are presented as mean ± standard deviation. Bars with asterisks differ significantly (* *p* ≤ 0.05; ** *p* ≤ 0.01).
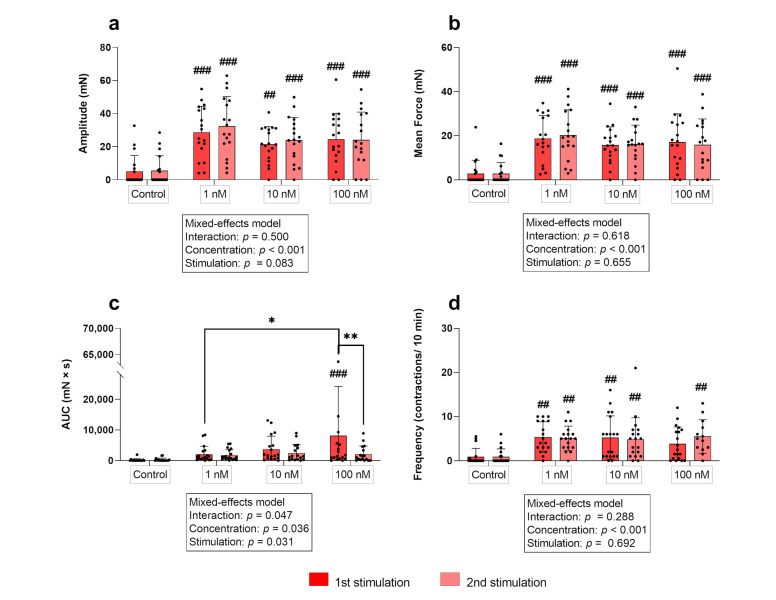
Figure 4(**a**) Amplitude, (**b**) mean force, (**c**) AUC, and (**d**) frequency of contractions of longitudinal myometrial strips stimulated with 1 nM, 10 nM, and 100 nM oxytocin. Responses following stimulation differed significantly from untreated controls (# *p* ≤0.05, ## *p* ≤ 0.01). The results are presented as mean ± standard deviation. Bars with asterisks differ significantly (* *p* ≤ 0.05; ** *p* ≤ 0.01; *** *p* ≤ 0.001).
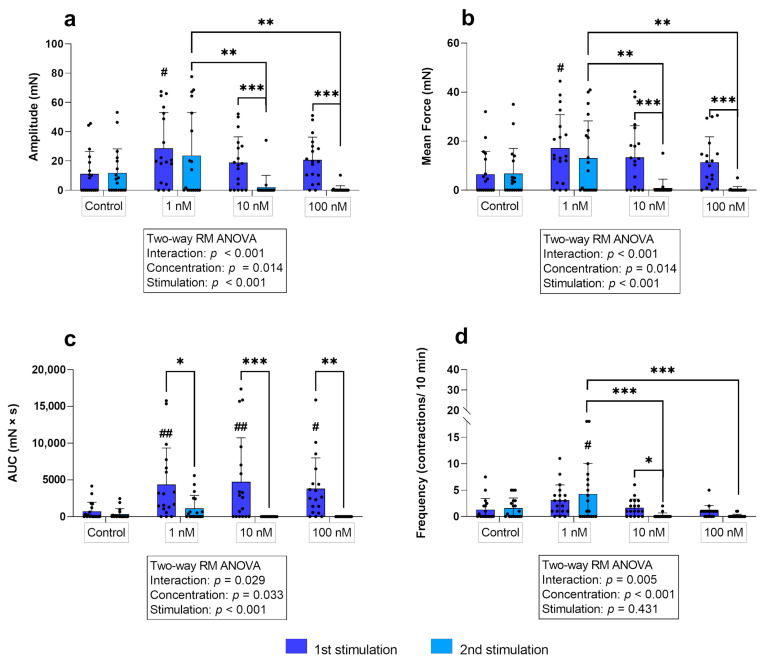

Figure 5Comparison of the circular and longitudinal myometrial layer responding to 1 nM, 10 nM, and 100 nM oxytocin during the first (1st) and second (2nd) stimulation with oxytocin. (**a**) Amplitude, (**b**) mean force, (**c**) AUC, and (**d**) frequency of contractions were analyzed and presented as mean ± standard deviation. Bars with asterisks differ significantly (** *p* ≤ 0.01; *** *p* ≤ 0.001).
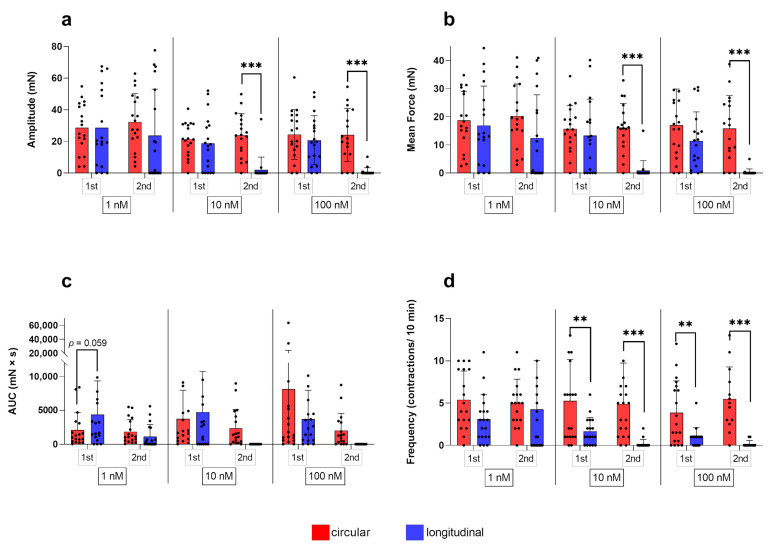



#### 3.1.3. Mean Force

In both layers, the effect of the oxytocin stimulation on the mean force of contractions was equivalent to that on the amplitude.

In the circular layer, the independent variables “concentration” and “stimulation cycle” did not interact. However, as with the amplitude, a main effect of the used oxytocin concentration was discovered (*p* ≤ 0.001), as the mean force of contractions increased significantly after each oxytocin stimulation compared to the control in both stimulations (*p* ≤ 0.001 for each; Figure 3b). The mean force of contractions after stimulation did not differ between oxytocin concentrations or when comparing the two stimulations cycles.

In the longitudinal layer, a significant statistical interaction (*p* ≤ 0.001) of the independent variables was found. Furthermore, simple main effects analysis showed a significant effect of the oxytocin concentration (*p* ≤ 0.05) and the stimulation cycle (*p* ≤ 0.001) on the mean force of contractions. No significant differences between untreated control and stimulated strips were found, except for the first-time use of 1 nM oxytocin (*p* = 0.014). The second application of 10/100 nM oxytocin caused a significantly reduced mean force compared to the effect at the first stimulation (*p* ≤ 0.001) and the second use of 1 nM oxytocin (Figure 4b).

As expected, when comparing the response of the layers, a significant interaction (*p* ≤ 0.01 for each) of variables was found for all oxytocin concentrations. Furthermore, for 10 nM and 100 nM, but not for 1 nM oxytocin a statistically significant main effect of “layer” (*p* ≤ 0.01 for each) and “stimulation cycle” (*p* ≤ 0.001 for each) was identified (Figure 5b).

#### 3.1.4. AUC

ANOVA/mixed-effects model analysis revealed interactions of “concentration” and “stimulation cycle” in both layers (*p* ≤ 0.05 for each). Additionally, in the two layers, simple main effect analysis showed a statistically significant effect of each factor on the AUC of contractions (concentration: *p* ≤ 0.05 for each; stimulation cycle: circular *p* ≤ 0.05/longitudinal *p* ≤ 0.001).

At the first stimulation, the AUC in the circular layer only differed significantly from the respective control when 100 nM of oxytocin was used (*p* ≤ 0.001) and only at the first stimulation (Figure 3c). Furthermore, when comparing the responses to 1 nM and 100 nM oxytocin, a significant difference was found (*p* = 0.019) (Figure 3c). Moreover, the AUC was significantly greater in the first compared to the second stimulation for each oxytocin concentration (Figure 3c).

Interestingly, in the longitudinal layer, all concentrations caused a significantly greater AUC compared to the untreated control at the first stimulation (*p* ≤ 0.05 for each), but a significant decrease occurred at the second stimulation (*p* ≤ 0.05 for each; Figure 4c).

At the second stimulation, the AUC differed neither from the respective controls, nor between the different concentrations in both layers (Figure 3c and Figure 4c).

In direct comparison of the layers, simple main effects analysis showed that the stimulation cycle had a statistically significant effect on the stimulating effect of all oxytocin concentrations on the AUC (*p* ≤ 0.01 for each), as the response was always stronger at the first application. Although AUC did not differ significantly between layers, it is noteworthy that using 1 nM and 10 nM oxytocin at the first stimulation the AUC was greater in the longitudinal than the circular layer, with even a trend to significance at 1 nM (*p* = 0.059; Figure 5c).

#### 3.1.5. Frequency

In the circular myometrial layer, no interaction of factors was found, but a significant effect of the “concentration” on the frequency of contractions (*p* ≤ 0.001). At the first stimulation, the frequency of contractions increased significantly following stimulation with 1 nM and 10 nM oxytocin (*p* ≤ 0.01 for each). This response was not only reproducible at the second stimulation, but all three concentrations also accelerated contractions significantly compared to the untreated control (Figure 3d).

Interaction of factors was observed in the longitudinal layer (*p* ≤ 0.05). As already described for the amplitude and the mean force of contractions, when stimulated for the second time, the response to 1 nM oxytocin, accelerating the frequency of contractions, was significantly stronger than the responses following the two higher concentrations in the longitudinal layer (*p* ≤ 0.001 for each; Figure 4d). Consequently, comparing the frequency of contractions after each stimulation revealed that 10 nM oxytocin elicited a significantly reduced frequency of contractions when administered for the second time (*p* ≤ 0.01; Figure 4d). However, as 100 nM usually triggered only one tonic contraction, the difference to the response in the second stimulation was not significant.

A main effect of the factor “layer” (*p* ≤ 0.001) was found and, accordingly, when comparing the response of layers, a significantly lower frequency in the longitudinal layer was observed in both stimulation cycles using 10 nM and 100 nM oxytocin (first stimulation: *p* ≤ 0.01; second stimulation: *p* ≤ 0.001; Figure 5d).

#### 3.1.6. Analysis of the First Oxytocin Induced Contraction

When evaluating the first contraction only, the initial response to oxytocin, results differed only slightly. Where previously only a trend was apparent, the variable “stimulation cycle” now influenced the amplitude in the circular layer (*p* = 0.016). Furthermore, in the longitudinal layer, the amplitude following stimulation with 1 nM oxytocin was now significantly higher compared to the control (*p* = 0.05). Additionally, the mean force in the longitudinal layer was not only significantly higher than the untreated control after 1 nM oxytocin (*p* ≤ 0.001), but also after 10 nM (*p* < 0.05). A significant difference in the mean force between both stimulations with 1 nM oxytocin was found for the longitudinal layer (*p* ≤ 0.05).

For the AUC, we identified more pronounced differences between the two techniques of data analysis (Figure 6). We found a significant increase in the AUC in the circular layer compared to the control when 10 nM oxytocin was applied (*p* ≤ 0.05). When looking at the effects of the different oxytocin concentrations separately, a trend to significance when comparing the AUC between layers after the second dose of 10 nM emerged (Figure 6). As already seen as a clear trend when comparing the average values, the AUC was significantly greater in the circular layer compared to the longitudinal one, responding to 1 nM oxytocin at first stimulation (*p* ≤ 0.01, Figure 6).

### 3.2. Denaverine

An example of the experimental design to test the effect of denaverine combined with oxytocin is given in Figure 2. In both myometrial layers, the parameters of amplitude, mean force, AUC, and frequency of contractions did not differ between control strips (stimulated with 1 nM oxytocin only) and strips receiving oxytocin directly followed by 1.97 nM or 2.97 nM denaverine, respectively (Figure 7). In both layers, adding 2.97 nM denaverine alone showed no change in the observed parameters in direct comparison to the time frame before application (Figure 2).

Prior priming of the tissue with 2.97 nM denaverine for 20 min did not impact the response to subsequent oxytocin stimulation. In both layers, the results for the analyzed parameters did not differ significantly between both treatments (Figure 8).

## 4. Discussion

In vitro techniques, such as the organ bath, are standard in drug testing and have been widely used to investigate the myometrial response to oxytocin stimulation using human tissue [48,49,50,51,52] and laboratory rodent tissue [53,54,55,56]. However, so far, only Gogny et al. [47] have studied canine myometrium in vitro. While their study was performed on uterine tissue samples from bitches in various stages of the estrus cycle, our study is the first to investigate the contractility of canine pregnant myometrium at term and its response to oxytocin stimulation in an in vitro set up.

As differences in the contractile in vitro myometrial responsiveness to oxytocin have been described in the pig depending on the uterine localization [57], all but two of our samples were taken from a similar location of the uterine horns to increase comparability. The uterine horns have been described to be more sensitive to oxytocin than the uterine body and the cervix, presumably allowing for a caudally directed pressure gradient to expel the offspring [57]. Interestingly, results from all the samples in the organ bath were comparable. Consequently, we decided to also include both uterine body samples in the data analysis.

To gain deeper insights, both myometrial layers were carefully separated and their contractile responses were evaluated individually. This separation of myometrial layers has not been previously described for dogs, but it was performed with guinea pig uterine tissue [58,59,60]. In human myometrium, the “mesh”-like microstructure [61] does not allow for separation, thus, explaining the lack of functional studies on separate myometrial layers in women so far [62]. Accordingly, in organ bath studies, the human myometrium is cut along the muscle fibers running parallel to the longitudinal axis of the uterus [49,52,63,64]. Even though in most mammalian animal species the myometrium consists of a well-defined circular and longitudinal layer, these are often not separated from each other for experiments. Tissue strips are simply cut in the corresponding fiber directions and mounted accordingly in the organ bath, so that the tissue strips still contain both layers, but only the contractions of the layer clamped in fiber direction can be measured [47,65,66]. In our experiments, separating canine myometrial layers was easy in all samples from the uterine horns and still possible in the two uterine body-derived samples. Although it is beneficial, separating the myometrial layers does not help in assessing whether inter-myometrial layer communication is a possible cause of uterine inertia. However, communication between the two myometrial layers via, for example, gap junction proteins, such as connexins, predominately Cx43 [67,68,69], is crucial for properly coordinated contractions in vivo [70]. Despite this, the organ bath was confirmed to be a suitable method to study the individual responsiveness of the layers to pharmacological treatment. Nevertheless, studying myometrial tissue stripes in toto might be an option for a better understanding of uterine inertia.

We identified a clear concentration-dependent response of the pregnant canine myometrium to oxytocin, as well as consistent differences between the distinct myometrial layers. To better illustrate our specific raw data, data analysis was conducted in two ways, using conventional mean values [49], followed by another analysis of only the “initial response”. This initial response to oxytocin appeared as a prolonged spastic constriction, especially after administering high oxytocin concentrations. Both methods provided comparable results and only pre-existing trends were reinforced to a significant difference when looking at the “initial response” evaluation, thus, supporting the suitability of both evaluation approaches.

Whereas 1 nM oxytocin caused regular contractions in both myometrial layers, 10 nM and 100 nM oxytocin induced one spasm-like contraction lasting several minutes, with subsequent suprabasal tension, with no further contractions being inducible in the longitudinal layer. In the circular layer, 100 nM and in some instances 10 nM oxytocin elicited a long-lasting tonic contraction as well. However, this was followed by a defined return to basic tension and subsequent regular, strong, but shorter contractions. The observation that high oxytocin concentrations induce a spasmogenic response with long-lasting tonic contractions is well described for isolated rat uteri at term (2 and 20 nM oxytocin) [71], and also for human pregnant myometrium cut in longitudinal fiber orientation (>0.5 nM) [49,72,73,74].

Differences between the circular and longitudinal myometrial layers with regard to different innervation and responsiveness to various stimuli have been reported in several species, such as cattle [75,76], pig [57,77,78,79], rat [80], rabbit [81], and guinea pig [60]. These differences are often attributed to the different ontogenetic origin of the layers [58,80]. While the circular layer evolves from the primitive muscles of the paramesonephric ducts, the longitudinal layer originates from subserosal connective tissue [82]. Comparable results for the myometrium of pregnant dogs are missing, and the stronger and more homogenous response in the circular myometrial layer compared to the longitudinal layer observed in this study differs from Gogny et al. [47]. Whether this is related to the reproductive status *per se* (non-pregnant vs. pregnant) requires further investigation. In addition to these postulated differences between the cyclic and pregnant myometrium, Crankshaw et al. [83] further described a steady oxytocin sensitivity of the longitudinal layer during the entire pregnancy and parturition in the rat. In contrast, a changing sensitivity of the former refractory circular layer becoming responsive to oxytocin towards parturition was found, indicating possible pregnancy-associated and/or periparturient changes in the myometrium in the dog, which deserve further research.

Nevertheless, the markedly reduced responsiveness of the longitudinal myometrial layer after stimulation with high doses of oxytocin is a new finding in canines. In human obstetrics, exposure to high doses of oxytocin during labor augmentation is associated with uterine atony and consecutive risk of severe post-partum bleeding [84,85]. In addition, prior use of oxytocin for labor augmentation affects the uterine response to subsequent oxytocin administrations. If another oxytocin injection was given to induce effective uterine contractions after medically indicated C-section (labor arrest), a nine-fold greater dose of oxytocin was required to prevent post-partum bleeding [86]. Similarly, in pregnant human myometrial fibers cut in a longitudinal direction, pre-treatment with oxytocin induced an attenuated response to increasing oxytocin concentrations in a concentration- and time-dependent manner when analyzing the amplitude, frequency, and AUC of contraction [63]. Those findings were attributed to the well described desensitization of G-protein-coupled receptors (GPCR) [87,88,89], such as OXTR [90,91]. It seems probable that desensitization of OXTR is causative for our finding in the canine longitudinal myometrium, too. Desensitization is a physiological process of GPCRs to prevent overstimulation by its substrate, leading to temporary or permanent inactivity after repeated stimulation [92]. Regarding OXTR, extensive oxytocin treatment significantly reduces oxytocin–OXTR binding and OXTR responsiveness in myometrial cells, diminishing the effect of subsequent oxytocin administration and, thus, hindering progress of labor [63,93,94,95]. Rajagopal et al. [92] distinguished between short-term transducer uncoupling and long-term downregulation of GPCRs. When activated by their respective substrate or an agonist, the GPCR undergoes phosphorylation within seconds to minutes. Regulatory proteins, β-arretins, bind to the phosphorylated receptor, uncoupling it from the G-protein. This represents acute desensitization of second messenger signaling. Additional ubiquitin-dependent internalization, leading to protein degradation and, thus, decreased receptor expression at the plasma membrane, occurs after long-term agonist exposition. If not degraded, the receptor is recycled by dephosphorylation in the endocytic compartment, which is referred to as resensitization [89]. With respect to our experiments, the lack of response to 10 nM and/or 100 nM oxytocin, especially in the longitudinal layer, is probably associated with OXTR desensitization. Given the shortness of the exposure duration, short-term desensitization with receptor uncoupling [92] seems to play a role in most of the organ bath experiments.

It remains to be clarified why only the longitudinal layer ceased to respond to oxytocin stimulation. Interestingly, we recently identified a significantly reduced OXTR protein expression in the longitudinal compared to the circular myometrial layer of bitches at term [21], supporting different roles and responsiveness of both myometrial layers. Whether this observation is related to a stronger or earlier desensitization in the respective layer requires further investigation. Strikingly, when investigating oxytocin-induced contractility in non-pregnant porcine myometrium in vitro, Kitazawa et al. [57] found a greater response of the longitudinal layer to low oxytocin concentrations (0.3–1 nM), whereas higher concentrations (1–3 nM) were needed to trigger contractions in the circular layer. Those findings suggest that lower oxytocin concentrations are more effective at inducing efficient myometrial contractions in the longitudinal layer in the pig. Images illustrating typical concentration-dependent oxytocin-induced contractions show an increase in the duration of the initial contraction in the longitudinal layer, with increasing oxytocin concentrations [57] quite similar to our findings. Additionally, in isolated pregnant rat uteri, low concentrations of oxytocin (0.2 nM) enhanced the force and frequency of phasic contractions, whereas higher concentrations produced a prolonged muscle contracture as discussed above [71,96]. This naturally raises the question of whether even lower oxytocin concentrations in organ bath experiments might have a more positive effect on contractility in the longitudinal layer, presenting an approach for further experiments.

Regarding the clinical application of oxytocin, our results suggest that only low concentrations of oxytocin are suitable for (re)activating myometrial contractions in the (peri)parturient dystocia bitch. High concentrations might cause spasm-like tonic contractions of both myometrial layers, obviously not promoting labor, but reducing fetal oxygen supply and thereby threatening fetal vitality and outcome [43]. Furthermore, subsequent contractile failure and atony of the longitudinal layer presumably hinders the effective progress of labor, thus, increasing the need for an emergency C-section. In addition to oxytocin, denaverine, currently registered for use in cattle only, was postulated to be beneficial in canine dystocia [31,97]. While in cattle some novel research exists addressing the effect of denaverine during labor [98,99,100,101], there are currently no studies investigating its functional effect on canine myometrium, or its possible clinical benefits. Using the organ bath approach, we failed to identify an effect of denaverine on myometrial contractions in the parturient bitch, neither alone nor combined with oxytocin. In vitro, we could not confirm the tonus reduction in oxytocin-induced spasmodic contractions, previously described in the isolated rat uterus, thereby increasing the efficiency of labor [102], as denaverine neither contributed to a decrease in the mean force nor induced a return to rhythmic, effective contractions. Furthermore, no priming or spasmolytic effect of denaverine before oxytocin stimulation was identified. This contrasts to the postulate that the combined use of these drugs is superior to oxytocin alone in reducing puppy loss during obstetric manipulation [31]. In cattle, a recent study could not identify a positive effect of denaverine on the birth process either [98]. Thus, the actual benefit of clinical use of denaverine in association with canine parturition or even uterine inertia remains questionable.

A limitation of the study is the inhomogeneity of specimens included in terms of breed, age, body weight, litter size, serum P4 concentration, and medical indication for the C-section. However, on the other hand, samples represent the entirety of the animals presented in clinics. Additionally, the time between sample collection and start of the experiment differed, mainly for logistical reasons. Nevertheless, our statistical analysis disproved an influence of age of bitches, serum P4 concentrations, and sample storage time on the investigated parameters, supporting the suitability of the results and relevance of data.

## 5. Conclusions

In summary, the organ bath technique is suitable for studying the contractile response of uterotonic substances on the isolated canine periparturient myometrium, namely the longitudinal and circular layer separately. Our results confirm different responses depending on the oxytocin concentrations used. Based on our findings, high doses of oxytocin, as often applied by some practitioners, contrary to the current recommendations, are not beneficial in dystocia management and should be avoided. Additionally, for the first time, we identified differences in the contractility of the longitudinal and circular myometrial layers, indicating different oxytocin sensitivity. All oxytocin concentrations induced or amplified existing contractility in the circular myometrial layer. The longitudinal layer showed an initial tonic contraction but did not respond subsequently when re-treated with higher doses of oxytocin (10 nM, 100 nM). This effect might be attributable to a short-term desensitization of the OXTR, uncoupling it from the respective G-protein. Further investigations into the distinct differences between the myometrial layers are crucial for a better understanding of canine parturition, not only during normal parturition, but also during dystocia due to uterine inertia.

## Figures and Tables

**Figure 1 biology-12-00860-f001:**
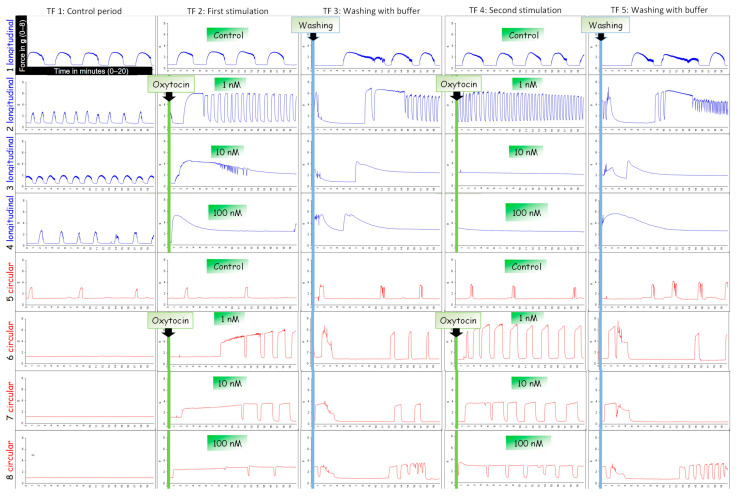
An example of typical contraction patterns of isolated canine myometrium. Four strips of the longitudinal (blue) and circular (red) layer were mounted in the organ bath recording spontaneous contractions (TF1). (TF2) Strips were stimulated with different concentrations of oxytocin (line 2/6: 100 nM; line 3/7: 10 nM; line 4/8: 1 nM) and afterwards washed with buffer (TF3). This stimulation cycle was repeated (TF4).

**Figure 2 biology-12-00860-f002:**
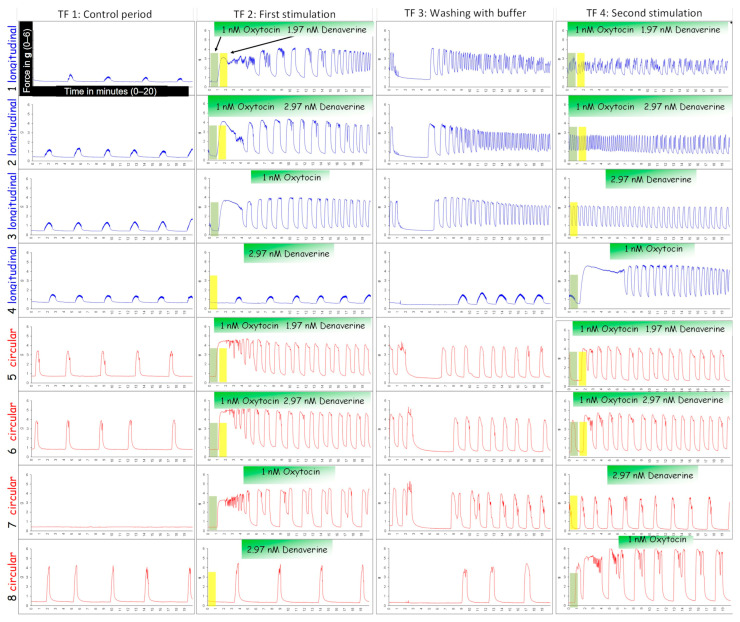
An example of an experiment investigating the effect of denaverine hydrochloride on canine pregnant myometrium in vitro. Strips of the longitudinal (blue) and circular (red) layer were stimulated with 1 nM oxytocin, followed one min later by the addition of either 1.97 nM denaverine (TF2; line 1/5) or 2.97 nM denaverine (TF2; line 2/6). One strip per layer was exposed to only 1 nM oxytocin (TF2; line 3/7) or 2.97 nM denaverine (TF2; line 4/8). Strips treated with denaverine only in TF2 were stimulated with oxytocin in the second stimulation cycle (TF4; line 4/8).

**Figure 6 biology-12-00860-f006:**
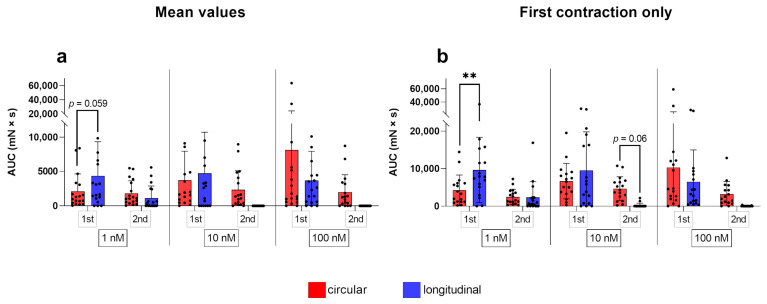
Analysis of up to 10 contractions (**a**) during the first (1st) and second (2nd) stimulation with oxytocin gave slightly different AUC results than considering only the initial response to oxytocin (**b**). The results are presented as mean ± standard deviation. Bars with asterisks differ significantly (** *p* ≤ 0.01).

**Figure 7 biology-12-00860-f007:**
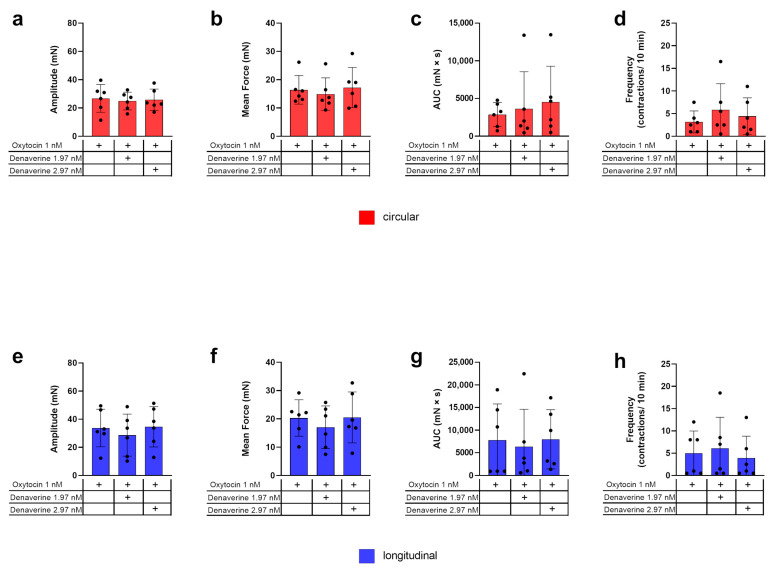
Investigation of the combined effect of oxytocin and denaverine and oxytocin only on the amplitude (**a**,**e**), mean force (**b**,**f**), AUC (**c**,**g**), and frequency (**d**,**h**) of contractions for the two myometrial layers, respectively (red = circular; blue = longitudinal). + indicates what has been applied to the samples. No statistically significant differences were found. The values are presented as mean ± standard deviation.

**Figure 8 biology-12-00860-f008:**
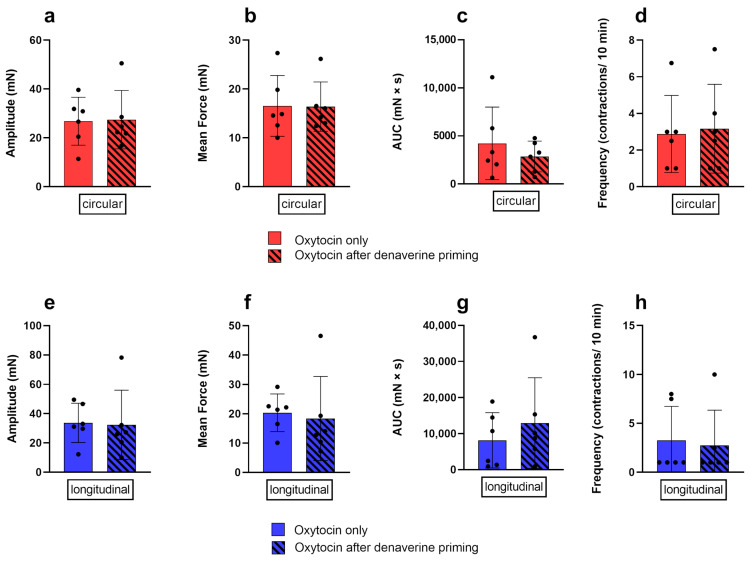
Comparison of the response of the circular (red) and longitudinal (blue) myometrial layer to oxytocin alone (no pattern) and after prior priming with 2.97 nM denaverine (black stripes). No differences between the treatments were found for the amplitude (**a**,**e**), mean force (**b**,**f**), AUC (**c**,**g**), and frequency (**d**,**h**) of contractions. Results are presented as mean ± standard deviation.

## Data Availability

The data presented in this study are available on request from the corresponding author.

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
