# Peer review of "The In Vitro Contractile Response of Canine Pregnant Myometrium to Oxytocin and Denaverine Hydrochloride"

_biology, 2023, doi:10.3390/biology12060860_

Round 1

Reviewer 1 Report

MDPI (biology- 2405965)

The ex vivo contractile response of canine pregnant 2 myometrium to oxytocin and denaverine hydrochloride

This work is conducted to evaluation a useless knowledge about the effect of oxytocin during labor to improve the medical care of dogs. I hope that the obtained results and conclusions will a helpful data as a background of research for the study the animal reproduction. Hence, the study is close to the scope of this journal and it can be widely cited. However, this is only a far observation, without description of molecular mechanism.

Additionally, I believe that the attached suggestions can help the authors to improve their manuscript.

Title/text: Please consider to change the term “ex vivo” on “in vitro”, since the circular and longitudinal muscle of myometrial strips were separated.

Line 31: I know that it was average amplitude, but what does it mean exactly? It can be amplitude from the lowest and the highest points of the force of contractions as generally. But amplitude can be also a sum of differences between force at points of measures one by one.  It has to be clarified. I saw many contractions arrows. For example, Fig. 1. Strips 6. After washing (TF3). The amplitude can be 3 (4-1)  during 12-18 (time), but it can be also 0.5 (4-3.5) at 17. It was only as example.    

Line 32: Frequency? How can you described the pick of contractions? Fig.1: 10 of picks at 1 longitudinal control. But what about 6 circular TF3 at 16-18 (time):   It was only as example.    

Lines 39-40: It did not affected or it was not studied?

Line 131: What months/parts of pregnant?

Line 141: Why are you sure that storage time had not impact on the response? What kind of gas (air/oxygen) was supplemented? It was a positive control for contractions / viability of strips?

Line 155: And what for the medium was changed after each 10 minutes?  

Line 159: Only “after” equalibration?

Fig.1.:

-          There is a different contractions at control period for longitudinal muscle 2 and 4 as well as 1 and 3.

-          After TF2 the control is without oxytocin? So, the arrows and Oxytocin has to go down. The same 5 circular – there was not oxytocin stimulation.

-          The order of doses has to be changed on Fig.1 , control, 1, 10, 100.

-          What was done after washing at TF3 for longitudinal 1? There was not oxytocin but it was an increase of contractions. Why the picks of contractions were disappear?

Line 491: Please consider, if it has to be emphasized (at Introduction) if the circular and longitudinal layer are separated at canine uterus in vivo?

Line 512: Since 1nM oxytocin caused (regular contraction) different effect like 10-100 nM (deregulated contractions), in previous study it has to be verification of the border dose.

Line 524: Chow & Marshal, 1981 without numbers.

Discussion/Conclusion. What is your opinion? How the different effect of different doses oxytocin on circular and longitudinal layer can finally regulate the contraction of uterus in vivo? What about the open/close the uterine horns? It has to be emphasized.

Line 54: “…suggest involvement [insert: of oxytocin] in prostaglandin…”

Author Response

Dear Reviewer,

Please see the attached letter.

Reviewer 2 Report

The manuscript from Jungmann et al investigates ex vivo effects of oxytocin and denaverine hydrochloride in the contractile activity of canine endometrium. This is a well-designed and innovative study, presenting high physiological and clinical relevance. In addition, the used methodology is novel for its application in the dog and opens the door to further interesting and relevant studies. However, this reviewer believes that there are some issues in the present work that should be solved before considering acceptance of the manuscript.

Abstracts: Overall, both the simple summary and abstract address the main topics and findings of the manuscript. As some small suggestions for improvement:

1.     the information in lines 24/25 and 40/41 referring to denaverine hydrochloride should be moved to the “results” part of the abstract (maybe right at the begin of this section) so that the abstract end with main message of the paper (oxytocin usage), keeping a more consistent structure of the text. 

2.     Line 21: remove the “highly interesting” to keep the scientific soundness of the text.

Introduction:

The introduction highlights the clinical relevance and knowledge gap in the field. Also, the physiological/pharmacological actions of the studied factors are presented.

As a general comment, contrasting with other sections of the paper, the introduction is not so easy to read. This reviewer is not a native English speaker, but it appears that several grammar issues are present mainly in the introduction section, that could lead to this issue. Also, several sentences are quite long, introducing a lot of information at once that makes them easily convoluted.  This reviewer would suggest some rephrasing, or even English editing, that would improve the manuscript quality, making the message more clear and the introduction more easy to read.Some identified examples:

Lines 47-49: long sentence, missing either commas or to be separated in two sentences.

Line 58: Remove “although”.

Lines 65-69: Long sentence, hard to follow, missing commas to separate ideas.

Line 72: “recently” is misplaced in the sentence

Additionally, in L. 65-69: This sentence is quite long, bringing a lot of information, but not being clear to message or physiological relevance of these changes. How were these differences? Decreased? Increased? Physiological relevance of RhoA/Rho?

Materials and methods:

All methods are described to a great detail and are easy to understand/follow. Additionally, different methodological options are later discussed in the discussion section.

Regarding samples collection:

-which were the medical reasons leading to the indication of c-section? Do these medical reasons have no impact on the observed results? Is it clear that inertia was not present in the used samples? Maybe a supplemental table with a list of medical conditions could be considered to make this more clear.

-regarding stage of pregnancy, there is no clear indication of late mid-gestation of actually at term, although this is stated later in line 471. Do the authors have access to P4 levels? If not, “term” should be replaced by “late pregnancy” in line 471. Furthermore, these limitations should be discussed in the discussion section.

-lines 142-143: How was this confirmed? Preliminary experiments? Observations during the experiments? Please give some details on this point.

Also, could this not lead to the lack of spontaneous contractions described in point 3.1.1? If not, what would be the authors’ explanation for this?

-Figures 1 and 2 are results! For this reason, they should be moved to the results section of the manuscript. Furthermore, I suggest changing the description to “representative results” of measurements instead of “example”.

Results

All the obtained results are described in high detail. The separation by amplitude/mean force/AUC/frequency make the description a bit long, but ok to follow.

-Reference to Fig 3 is missing in line 306.

-line 450 – the expression “quite similar” is more for discussion section. I would rather highlight in the results section the lack of significant effects/differences.

-Figure 7 – the letter for individual graphics are at the bottom left, whereas in all other graphics in the paper are at the top left corner of each graphic. Not sure if this is by design or was overlooked, but all letters on the top left corner would keep consistency throughout the manuscript.

-In this reviewer’s opinion, some figures could be combined into one. This is the case for the figures 3 and 4, and the figures 7 and 8. This because the figures address similar experiments and/or analysis or topic. This would make easier for the reader to find the figure with the results described. In special for figures 3 and 4, and the sections 3.1.2-3.1.5 address both, making the reader constantly “jump” between them.

Discussion

The discussion is well written, addresses all the main question regarding the obtained results and their relevance, in addition to the limitations of the work.  

Main issues with english refer to the Introduction section. 
This reviewer would suggest some rephrasing, or even English editing, that would improve the manuscript quality, making the message more clear and the introduction more easy to read.

Author Response

Dear Reviewer,

Please read the attached letter. 

Reviewer 3 Report

Line 142: Storage time had no impact on the response of the tissue to subsequent oxytocin stimulation, please provide the appropriate references.

Line 179: How were the oxytocin concentrations chosen? Please provide references. Line 197: Also about the Denaverine hydrochloride (Sensiblex©).

Author Response

(The authors gave the same response as above.)

Reviewer 4 Report

The article is well written. The topic is relevant for practice. The material and method can also be used in other areas for uterine research. Only some questions in the text have to be answered or supplemented.

I have listed them below:

It should be better described why the cesarean section was necessary after the applications.

Did you take progesterone measurements before the operation? If yes the values should be written. If no, the day or days of the operation should be written or described (whether the time of the operations was the same for all bitches?).

The age of bitches is to be added. Was there a difference between the old and young bitches? This also applies to the number of puppies.

It should be discussed whether these factors can produce an influence or difference after application ?

Best regards

Author Response

(The authors gave the same response as above.)

Round 2

Reviewer 2 Report

In the revised manuscript, Jungmann and colleagues provided specific answers for the observations or suggestions from the first review, and proceeded with several corrections and improvements of the manuscript.

Overall, the quality of the text appears to have been improved, with a special focus on the introduction, where the message is more clear. Also, inconsistencies have been solved.

The added information and explanations provided regarding samples collection answers to the main concerns of this reviewer. And the addition of the supplemental table improved the clarity of the work. Despite still maintaining the opinion that Fig. 1 and 2 should be placed in the results section, this reviewer understands the point of view of the authors.

Also, other small issues throughout the manuscript were improved.

Thus, following all this, this reviewer supports the acceptance of the manuscript for publication on the current form.

If it is allowed to this reviewer to share an opinion, on a side note, the variability of animals used in the present study (age and breed), although frequently considered a “limitation” on experimental designs, can be considered also a strength in the present work. Effects on high homogenous samples frequently fail in their translation to the clinics/real world (not every significant result is important). Thus, by observing significant effects on such a group of animals heterogeneous between them, but more representative of the population that appears every day in our clinics, this author is of the opinion that there is a higher chance of clinical applicability of the insights obtained with the present work.